# Clinical validation of an open-access SARS-COV-2 antigen detection lateral flow assay, compared to commercially available assays

**Christine M. Bachman**◉*◔, **Benjamin D. Grant**◔, **Caitlin E. Anderson, Luis F. Alonzo, Spencer Garing, Sam A. Byrnes**¤, **Rafael Rivera, Stephen Burkot, Alexey Ball, James W. Stafford, Wenbo Wang**◉, **Dipayan Banik, Matthew D. Keller, David M. Cate**◉¤, **Kevin P. Nichols**¤, **Bernhard H. Weigl, Puneet Dewan**

Global Health Labs, Inc, Bellevue, Washington, United States of America

◔ These authors contributed equally to this work.
¤ Current address: Amazon Dx, Seattle, Washington, United States of America
* Christine.bachman@ghlabs.org

## Abstract

Rapid tests for SARS-COV-2 infection are important tools for pandemic control, but current rapid tests are based on proprietary designs and reagents. We report clinical validation results of an open-access lateral flow assay (OA-LFA) design using commercially available materials and reagents, along with RT-qPCR and commercially available comparators (BinaxNOW® and Sofia®). Adult patients with suspected COVID-19 based on clinical signs and symptoms, and with symptoms ≤7 days duration, underwent anterior nares (AN) sampling for the OA-LFA, Sofia®, BinaxNOW ™, and RT-qPCR, along with nasopharyngeal (NP) RT-qPCR. Results indicate a positive predictive agreement with NP sampling as 69% (60% -78%) OA-LFA, 74% (64% - 82%) Sofia®, and 82% (73% - 88%) BinaxNOW™. The implication for these results is that we provide an open-access LFA design that meets the minimum WHO target product profile for a rapid test, that virtually any diagnostic manufacturer could produce.

## Introduction

With more than 3,250,000 deaths worldwide [1] due to the Covid-19 pandemic, early detection of infectious patients is required to aid in efforts to interrupt transmission. One of the most effective tools to detect potentially infectious cases of COVID-19 are rapid diagnostic tests for the virus that causes the disease, SARS-COV-2. Transmission models have predicted early detection in the community coupled with isolation may accelerate reductions in transmission [2, 3].

At present, most health facilities rely on polymerase chain reaction (PCR) -based tests, which are based on amplification of genetic material for detection of the virus. While enormously sensitive, PCR poses several challenges. For example, time to results may be prolonged, cold chain may be required to transfer samples to lab facilities, expensive reagents are utilized,

**Data Availability Statement:** All relevant data are within the paper and its Supporting Information files.

**Funding:** All Authors of the manuscript work for the funding organization - Global Health Labs. https://www.ghlabs.org/.

**Competing interests:** No authors have competing interests.

**Abbreviations:** AN, Anterior Nares; AUC, Area under the Curve; C-PAP, Covid Technology Access Pool; FDA, Food and Drug Administration; LMIC, Low- and Middle-Income Countries; LFA, Lateral Flow Assay; MSD, Meso Scale Discovery; NP, Nasopharyngeal; OA-LFA, Open Access Lateral Flow Assay; PCR, Polymerase Chain Reaction; TPP, Target Product Profile; VTM, Viral Transport Medium; WHO, World Health Organization.

and trained lab technicians are needed to perform the testing [4]. Simpler and less-expensive antigen-based rapid tests have been developed to address these issues. While they may not be as sensitive as PCR, antigen-based rapid tests are generally inexpensive, amenable to mass-production, can be conducted by minimally trained individuals, and offer nearly immediate results for public health actions [5]. The Food and Drug Administration (FDA) has provided Emergency Use Authorizations (EUA) to (as of 5/17/2021) 24 rapid antigen diagnostic tests, most of which are based on detection of SARS-COV2 nucleocapsid protein [6] All of these assays are proprietary and are produced by commercial entities.

We previously published the development of a rapid test in the lateral flow assay (LFA) format to detect the nucleocapsid protein of the SARS-CoV-2 virus [7]. This assay is distinct from available commercial assays in that the assay design and architecture were released as open access, and that all reagents are commercially available. *In-vitro* performance of the assay met the World Health Organization (WHO) target product profile (TPP) for a rapid diagnostic test [8]. Specifically, the analytical sensitivity achieved the target WHO target of $10^6$ genomic copies/mL. Here we report the subsequent clinical validation procedures and results from this prototype lateral flow assay, in comparison to RT-qPCR results from swabs taken from nasopharyngeal and anterior nares sampling. Two additional experimental approaches were evaluated in this study–a mobile phone application to image and interpret a completed LFA, and a system to evaluate a patient's sense of smell. We include on-site rapid test results from two commercially available rapid tests as comparators, the Quidel® Sofia®SARS Antigen FIA and the Abbot® BinaxNOW™ COVID-19 Ag card to compare performance.

## Methods

### Study design

An open-label prospective study design of adults aged 18 and above presenting with symptoms of Covid-19 in 2 private clinics in Los Angeles County, California. These sites offer drive-through testing services as a part of standard clinical care. Ethics approval was obtained by IntegReview IRB (GHLPOC-01), USA. The IRB committee approved of consent obtained electronically.

### Participants

Eligibility criteria include symptomatic adult patients (age ≥18 years) with an onset of symptoms within 7 days. Symptoms included at least one of the following: fever, cough, shortness of breath, fatigue, muscle or body aches, loss of taste or smell, headache, congestion of nose, sore throat, nausea, vomiting, or diarrhea. Participants were consecutively offered voluntary enrollment in the study.

### Point-of-care test methods

The index test being evaluated is the OA-LFA. Briefly, the OA-LFA is a rapid diagnostic test with time to results in 30 minutes and uses anterior nares (AN) samples. Comparator point-of-care (POC) tests were also used to evaluate against the OA-LFA performance. These tests included the Sofia® SARS Antigen Fluorescent Immunoassay (FIA) (Quidel Corporation, San Diego, California, US) and the BinaxNOW™ Rapid Antigen Test for SARS-CoV2 (Abbott, Abbott Park, Illinois, US). One nasopharyngeal swab (NP) and four AN swabs were taken per patient. The NP swab was collected for reference RT-qPCR testing, and one AN swab was collected for each POC test. An additional AN swab was collected for RT-qPCR and reference antigen-testing. The rapid test results were blinded between research staff.

The OA-LFA utilized a PurFlock Ultra Elongated swab (Puritan Medical Products, Guilford, Maine, US, catalog no. 25-3806-U BT) for AN sampling. The BinaxNOW™ and Sofia® tests utilized swabs provided with each respective kit. For each test, both nostrils were swabbed by a healthcare professional. The swab was inserted into the anterior nares portion of the nostril until resistance was met and rotated five times in one direction before removing and repeating the process on the other nostril. First, the Sofia® and OA-LFA tests were each swabbed in opposite nostrils to allow each test a sample from a previously unsampled nostril. Specifically, the OA-LFA swab was used in the right nostril and the Sofia® swab was used in the left nostril prior to sampling the other nostril. After samples were taken for both Sofia® and OA-LFA tests, the BinaxNOW™ swab and reference AN swab were performed in a similar manner. The three LFAs were tested immediately after collection. The reference AN swab, also a Puritan PurFlock swab, was placed back in its dry transport container and placed on ice. The NP swab (Puritan Medical Products, Guilford, Maine, US, catalog no. 25-3306-H) was placed into a 15 mL screw top tube. At the end of each day, the reference AN and NP swabs were transported to a -20˚C unit where they remained for up to two weeks before transport and storage at -80˚C.

Immediately after sampling, the OA-LFA swab was placed into a sample tube containing 500 μL of running buffer (2% IGEPAL CA-630 [Sigma, St Louis, MO, I8896] in 1X PBS [Thermo Fisher, Waltham, MA 10010023]). The swab was rotated in the tube three times and then allowed to sit for 1 minute, after which the swab was discarded. An exact volume transfer pipette (Electron Microscopy Science, 70969–29) was used to remove 150 μL from the sample tube and gently pipetted onto the sample port of the LFA. After 30 minutes, the OA-LFA was read visually followed by quantitation of the test line intensity via commercial LFA reader (AX-2X-S, Axxin, Australia). Line intensity was also quantified via a GH Labs developed mobile LFA reader application (described below). The Sofia® and BinaxNOW™ were run per kit insert without modifications.

## Reference method testing

The AN swab and NP swab were both used for reference RT-qPCR testing at GH Labs after being stored and shipped at –80˚C. Additionally, antigen levels were measured utilizing an electrochemiluminescence immunoassay on the Meso Scale Discovery (MSD) MESO QuickPlex SQ 120 (MSD, Rockville, Maryland).

## Nasopharyngeal swab rehydration

Swabs were removed from the –80˚C freezer and set to warm to room temperature for 10 minutes prior to rehydration. Then, 500 μL of Tris-EDTA buffer solution, pH 7.4 was added to NP swab for 10 minutes and incubated at room temperature. Swab was vortexed and buffer solution was removed from tube for RNA purification.

## Anterior nares swab rehydration

The AN swabs were placed in a tube containing 500 μL of Viral Transport Medium (VTM). The VTM was made using the Center for Disease Control Recipe sterile Hanks Balanced Salt Solution (1X HBSS with calcium and magnesium ions, no phenol red) with 2% FBS (sterile heat inactivated fetal bovine serum), 100μg/mL Gentamicin and 0.5μg/mL Amphotericin B final concentration). The swab was rotated in the VTM three times and then allowed to sit for 10 minutes, after which the swab was discarded. 81 μL of the sample was set aside for the MSD immunoassay and the remainder was handed off for RT-qPCR.

## PCR testing

The RT-qPCR uses the CDC 2019-nCoV Real-Time RT-PCR Diagnostic Panel (SOP: CDC-006-00006). RNA is extracted from nasal swab samples using the QIAamp Viral Mini Kit as recommended by the CDC. The extracted RNA is amplified using a multiplexed version of the CDC 2019-nCoV Real-Time RT-qPCR targets (three targets: N1, N2, and RP) with the QuantaBio qScript XLT 1-step mix on a BioRad CFX96 Real-Time PCR Detection System [9]. The N1 gene is used to quantify SARS-CoV-2 viral load and the RP gene is used to quantify human RNA load. The RT-qPCR standards are control plasmids for both SARS-CoV-2 and Human RP gene which are procured from ITD. Both control plasmids are quantified in-house using the BioRad Digital PCR system. Samples were considered positive with a CT value of <40 [10] A standard curve for the N1 gene is run on each plate and utilized to calculate the viral-load from the CT value [9]. The N1 gene is used to quantify SARS-CoV-2 viral load and the RP gene is used to quantify human RNA load. The RT-qPCR standards are control plasmids for both SARS-CoV-2 and Human RP gene which are procured from ITD. Both control plasmids are quantified in-house using the BioRad Digital PCR system. Samples were considered positive with a CT value of <40 [10] A standard curve for the N1 gene is run on each plate and utilized to calculate the viral load from the CT value. Scientists performing RT-qPCR were blinded to antigen test results obtained at the clinical site.

## Anterior nares antigen level testing

Nucleocapsid antigen levels were determined using a GH Labs developed Meso Scale Discovery (MSD) immunoassay. The sandwich assay utilizes two monoclonal antibodies, 40143-MM08 and 40143-MM05 (Sino Biologicals), that exhibit high specificity to SARS and SARS-CoV-2 nucleocapsid proteins [11]. Sample lysis is achieved utilizing Igepal CA-630 (Sigma I8896). In addition to Igepal, the lysis buffer contains 1.5 mg/mL HBR-1 (Heterophilic Blocking Reagent 1, Part 3KC533, Scantibodies Laboratory Inc., USA) to reduce background signal. Quantification was performed using a standard curve prepared with recombinantly produced SARS-CoV-2 nucleocapsid (Acro Biosystems, Nun-C5227). A cut-off point of 10 pg/mL was chosen as the threshold for MSD positivity. This value was chosen to provide a minimum of 95% negative predicative agreement against both NP and AN RT-qPCR.

## Mobile reader application

An Android application ("reader app") was developed to capture images of the completed LFA tests and to determine whether control and/or test lines could be detected in the images. The reader app was based on the "rdt-scan" framework originally developed by Park *et al.* [12] and licensed under an open-source BSD-3 clause. Several features were added to or modified from the original framework to customize the reader app for use in this study. As detailed further in (see S1 File), this included placing a sticker with two ArUco codes on each cassette (see S1 Fig) to provide visual features ("keypoints") for the app to recognize on the otherwise-blank cassettes, as well as modifying how the app determines which keypoints are the most reliable. Further updates included configuring the app to search for control and test lines in the proper locations, optimizing the preprocessing steps prior to peak detection, and determining the proper threshold to distinguish the test line from background noise after converting the cropped test strip region into 1-dimensional intensity profiles along the direction of flow, averaged across the width of the strip.

After the app was initiated, the user had a live video view from the camera, and the app ran several real-time analyses on each video frame that enable prompts to the user to adjust various factors related to the image acquisition (see S2 Fig). Once the app recognized that the live

image contained the designated LFA (by identifying the ArUco codes) and passed all quality checks, including a new check for flatness of the phone, it saved the current video frame for processing. The authors also added a trigger at this point to immediately acquire a new, high-resolution still image, which was used for results presented in this manuscript.

In this study, the reader app was used across four Moto G7 Power mobile phones to acquire images from 155 of the 170 GH Labs tests. Based on findings from pre-trial testing with spiked serial dilutions, the metric reported to the user for control and test lines was the prominence, or peak height above baseline, calculated from the 1D intensity profile of the red color channel extracted from the RGB mobile phone images (see S1 File for further details). If no such peak greater than a set threshold could be found in the pre-defined control or test line regions of the image, the app reported "no line found." For computing receiver operating characteristic (ROC) curves and concordance values against visual read and PCR results, the peak height above baseline threshold was adjusted to find the optimal cutoff point.

### Clinical symptoms and standardized anosmia assessment

Patients self-identified for signs and symptoms of COVID-19, including fever or chills, cough, shortness of breath or difficulty breathing, fatigue, muscle or body aches, headache loss of taste or smell, sore throat, congestion or runny nose, nausea, or vomiting, and/or diarrhea. Standardized assessment for anosmia was conducted in parallel. USMELLIT is an integrated card and mobile application to systematically assess patients for anosmia loss of smell [13]. Cards present a random sequence of scent challenges in a 'scratch and sniff' format. The health care provider administered a card to the patient and input the patient ID into the app after scanning the QR code on the card. Patients were requested to scratch 5 sections of the card and associate an object with every section they smelled. Examples of smells associated with the card include orange, banana, roses, mint, and no-scent. Patients were scored on the number of correct answers they received out of 5. This number was captured in the case report form later for data analysis.

### Analysis

Sensitivity and specificity were calculated for each POC test compared to AN and NP RT-qPCR. Exact binomial confidence intervals were calculated in R using epiR [14]. Sensitivity was also calculated for tests stratified by AN and NP RT-qPCR viral load. Differences between the sensitivity and specificity of each LFA relative to AN and NP RT-qPCR were assessed in R using McNemar's mid-P test [15].

## Results

In the month of December 2020, a total of 170 subjects were enrolled at two drive-in sites in Riverside and Orange County. As shown in **Fig 1**, 110 of the 170 individuals (64%) were positive by NP PCR and 92 (54%) were positive by AN PCR. The logarithmic value of NP and AN viral load showed strong correlation (Pearson's Correlation Coefficient of 0.897), as shown in **Fig 2**. All subjects positive by AN PCR were also positive by NP PCR. Expectedly, discordant results (negative AN PCR and positive NP PCR) occurred when NP PCR viral loads were low. Overall, 84% of NP PCR positive cases were also detected by AN PCR.

The sensitivity and specificity for the OA-LFA, Sofia®, BinaxNOW™ tests and the MSD antigen reference test relative to both NP and AN PCR are shown in **Table 1**. Overall and in this patient population, amongst the rapid tests, BinaxNOW™ showed the highest sensitivity, followed by Sofia® and OA-LFA. All three rapid tests met the WHO TPP acceptable sensitivity target of 80% relative to the AN PCR. Relative to the NP PCR, only the BinaxNOW™ met

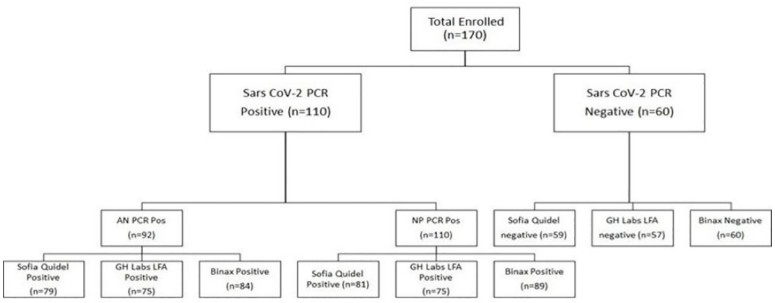

**Fig 1. Flowchart of samples positive and negative for study tests.**

this clinical sensitivity threshold, with a sensitivity of 82%. All of the rapid tests exceeded the WHO TPP acceptable specificity target of 97%, relative to the NP PCR.

We stratified the data based on a cutoff of 1000 copies/μL to understand the performance of each test at higher and lower viral loads. This cut-off corresponds to the acceptable analytical sensitivity outlined in the WHO TPP [8]. The results are shown in **Table 2**. In the lower viral load cases detected by NP PCR, all rapid tests demonstrated low sensitivity. Amongst the rapid tests, sensitivity among lower viral load cases was greatest for BinaxNOW™, but was still only

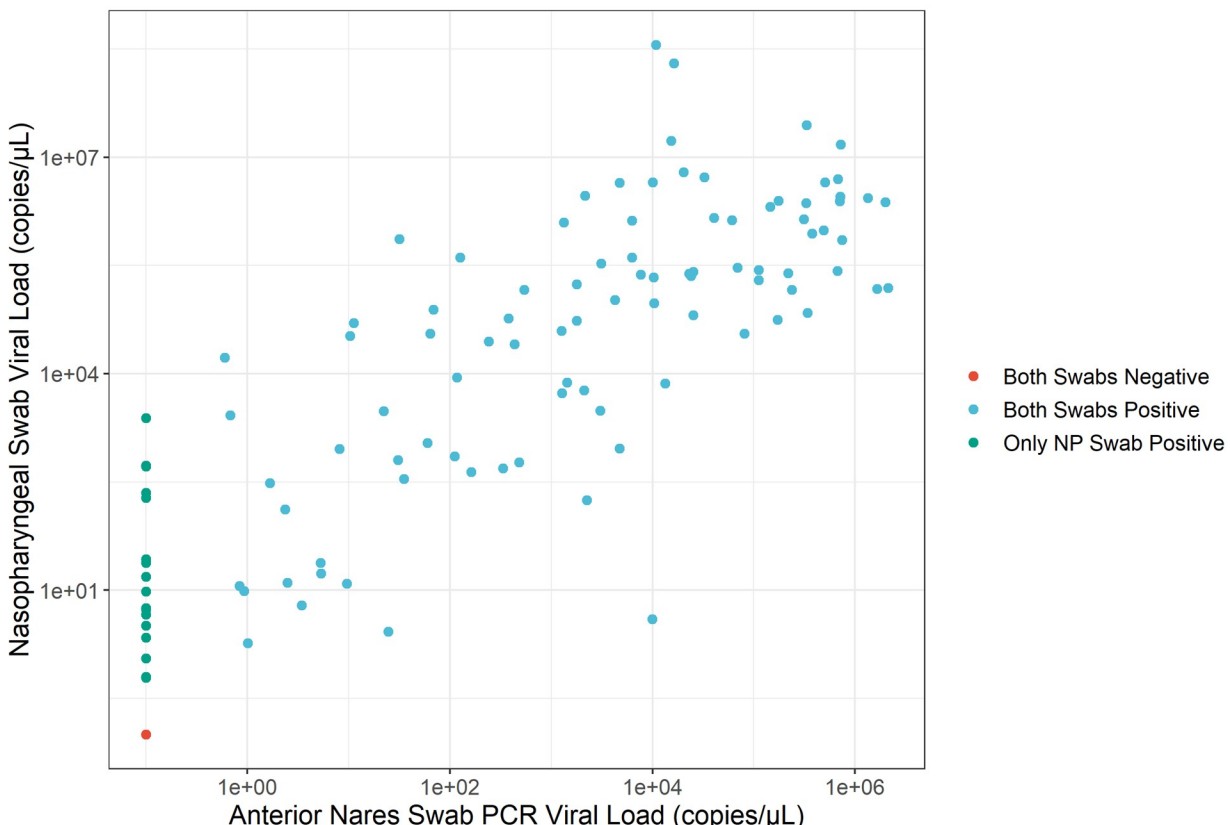

**Fig 2. Nasopharyngeal vs anterior nares PCR viral load results.** Cases where both swabs were positive are blue, both negative are red, and nasopharyngeal are in green. There is a clear positive correlation; however, the anterior nares swabs from the same individual have a lower viral load.

Table 1. Positive percent agreement and negative percent agreement values, including 95% confidence intervals, for all assays with either nasopharyngeal (NP) and anterior nares (AN) PCR as reference.

| Assay | NP Swab PCR | | AN Swab PCR | |
|---|---|---|---|---|
| | Sensitivity (95% CI) | Specificity (95% CI) | Sensitivity (95% CI) | Specificity (95% CI) |
| OA-LFA | 69% (60%-78%), 75/108 | 97% (88%-100%), 57/59 | 83% (74%-90%) 75/90 | 97% (91%-100%), 75/77 |
| Sofia® | 74% (64%-82%), 81/110 | 98% (91%-100%), 59/60 | 86% (77%-92%), 79/92 | 96% (89%-99%), 75/78 |
| BinaxNOW™ | 82% (73%-88%), 89/109 | 100% (94%-100%), 60/60 | 91% (84%-96%), 84/92 | 94% (85%-98%), 72/77 |
| MSD Ag ELISA | 79% (70%-86%), 87/110 | 98% (91%-100%), 59/60 | 91% (84%-96%), 84/92 | 95% (89%-99%), 74/78 |
| AN Swab PCR | 84% (75%-90%), 92/110 | 100% (94%-100%), 60/60 | NA | NA |

53%. In higher viral load cases, all rapid tests performed well, with sensitivity exceeding the 90% relative to both AN and NP reference PCR. In **Fig 3,** test results for PCR positive cases are shown in relation to AN and NP PCR viral loads. The WHO TPP acceptable analytical sensitivity target of 10^3 copies/μL is shown to highlight test performance above and below this threshold.

Using the McNemar mid-P value test for paired proportions, we compared the sensitivity and specificity between the three tests. Due to the small number of discordant results between the three tests it was determined that the mid-P variant of McNemar's test was the most appropriate [16] and results are shown in **S1 Table.** Pairwise comparison demonstrated that BinaxNOW™ had a significantly higher sensitivity (p < .05) relative to the OA-LFA or the Sofia® assays, using either AN or NP PCR reference. Amongst patients with AN PCR viral loads greater than 1000 copies/μL, there were no significant difference in sensitivity between any LFA. Although the study was not powered to detect statistical significance between specificities, the specificities of all three tests were similar and high.

As seen in **Fig 4A**, the reader app performance could largely mirror visual interpretation of the GH Labs test, with a single discordant case that appeared to stem from an improper visual interpretation. When compared against PCR reference values (Fig 4B and 4C), the reader app ROC curves produced AUC values of 0.95 and 0.88 versus AN swab and NP swab results, respectively. Optimal sensitivity and specificity from these ROC curves were 86% and 99% for AN swabs and 72% and 97% for NP swabs. Compared with visual performance (**Table 1**), the reader app produced equivalent to slightly higher (up to 3%) concordance values with PCR.

ROC curves were generated for the standardized anosmia assessment results versus PCR result (**S3 Fig**). The resulting area under the curves (AUC) are 0.69 relative to NP PCR and 0.63 relative to AN swab PCR. Classifying individuals as positive based on either a zero USMELLIT score or a positive LFA result increases the sensitivity with NP PCR, without

Table 2. Positive percent agreement, including 95% confidence intervals, for all assays with reference to both nasopharyngeal and anterior nares PCR.

| Assay | NP Swab PCR | | AN Swab PCR | |
|---|---|---|---|---|
| | Sensitivity < = 1000 copies/μL (95% CI) | Sensitivity >1000 copies/μL (95% CI) | Sensitivity < = 1000 copies/μL (95% CI) | Sensitivity >1000 copies/μL (95% CI) |
| OA-LFA | 25% (12%-42%), 9/36 | 92% (83%-97%), 66/72 | 55% (36%-73%), 17/31 | 98% (91%-100%), 58/59 |
| Sofia® | 38% (22%-55%), 14/37 | 92% (83%-97%), 67/73 | 64% (45%-80%), 21/33 | 98% (91%-100%), 58/59 |
| BinaxNOW™ | 53% (35%-70%), 19/36 | 96% (88%-99%), 70/73 | 79% (61%-91%), 26/33 | 98% (91%-100%), 58/59 |
| MSD Ag ELISA | 46% (29%-63%), 17/37 | 96% (88%-99%), 70/73 | 76% (58%-89%), 25/33 | 100% (94%-100%), 59/59 |
| AN Swab PCR | 57% (39%-73%), 21/37 | 97% (90%-100%), 71/73 | NA | NA |

Results are binned into two groups based on if the reference method measured less than or equal or above 1000 copies per microliter.

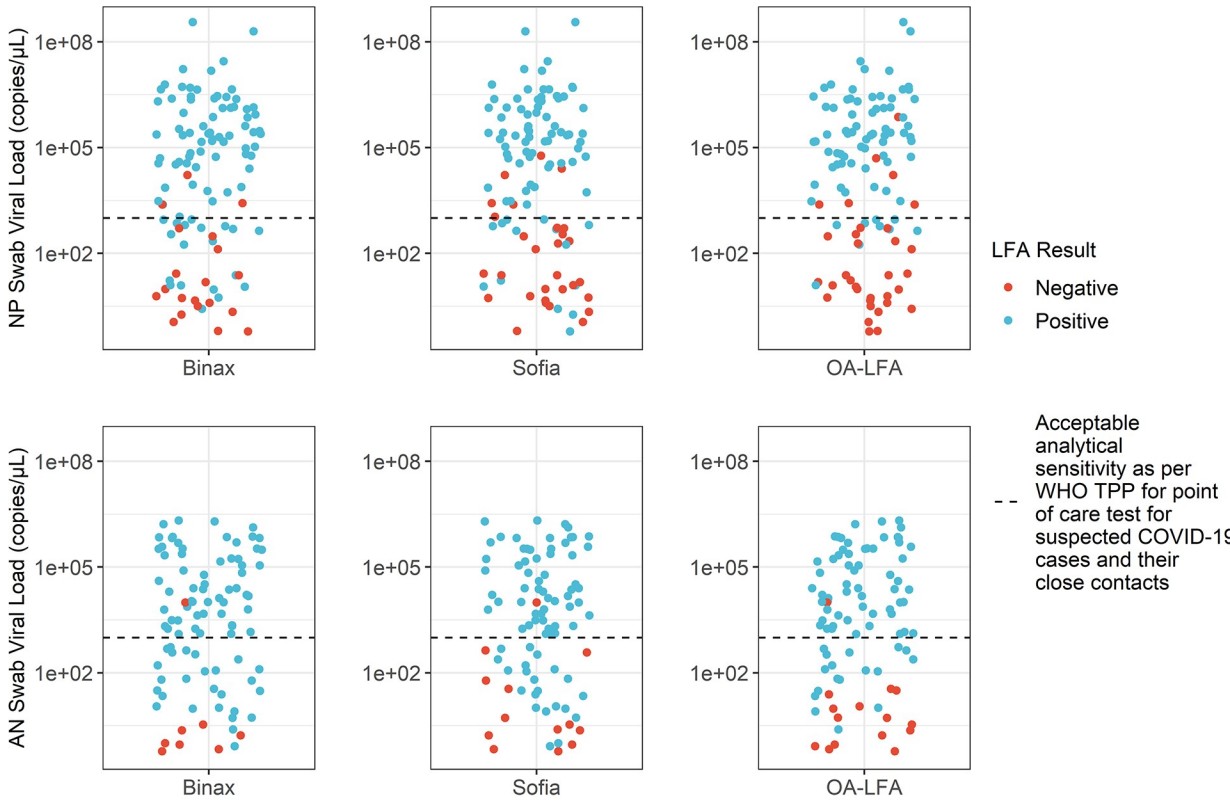

**Fig 3. Test results as a function of NP (top) and AN (bottom) PCR positive tests.** All tests perform well above the WHO TPP analytical sensitivity target of $10^3$ copies/µL.

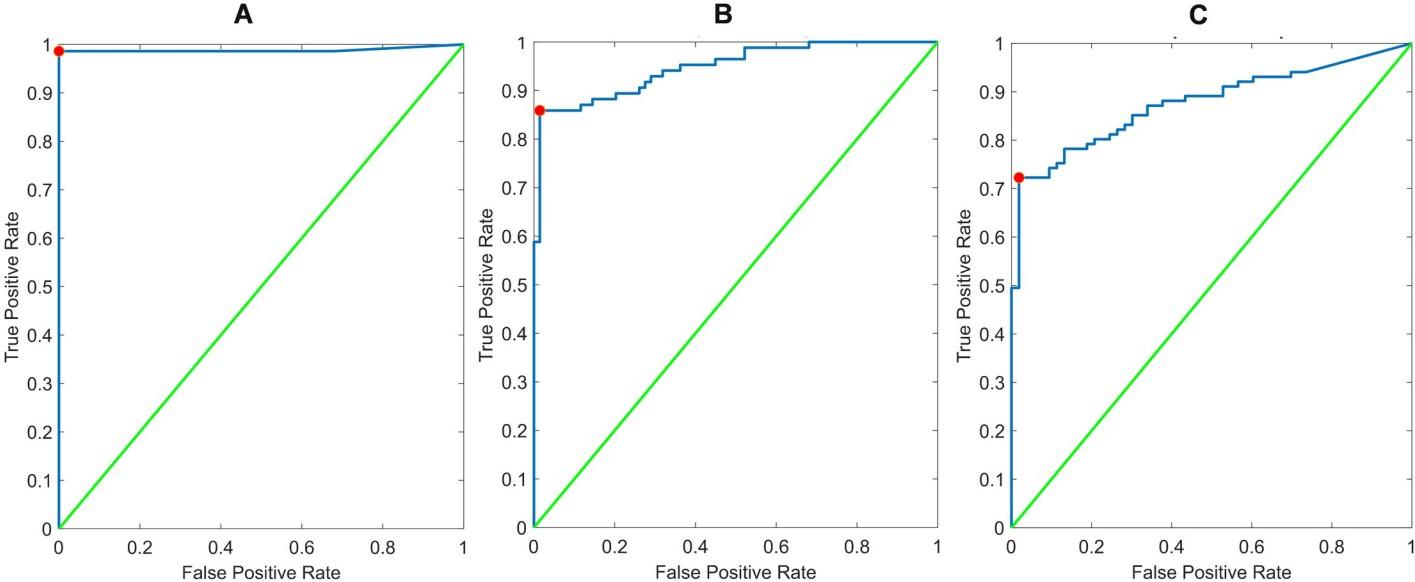

**Fig 4.** ROC curves for reader app performance versus (A) visual interpretation of GH labs assay, (B) AN swab PCR, and (C) NP swab PCR results. All represent use of the test line's peak height above background metric extracted from the red channel of the high-resolution images acquired by the reader app. AUC values were 0.99, 0.95, and 0.88 for panels A, B, and C, respectively.

affecting the specificity. Specifically, the sensitivity for the OA-LFA, Sofia® and BinaxNOW™ increases from 69% to 75%, 74% to 80% and 82% to 84%, respectively. Similar improvements were not seen relative to AN PCR. Sensitivity and specificity for each test in combination with USMELLIT is provided in **S2 Table**.

## Discussion

COVID-19 has created an unprecedented demand for rapid diagnostic testing to reduce transmission. Existing commercial tests have addressed some of the demand, yet wider use of more frequent testing will require innovative approaches to expand supply and access to effective diagnostics. Here we report the clinical validation results of an open-access LFA design that meets the minimum WHO analytic sensitivity requirements and operational characteristics for a point of care test for suspected COVID-19 cases and their close contacts, that virtually any diagnostic manufacturer could produce.

Clinical validation in this population of symptomatic adults, with duration of symptoms <7 days, shows that the OA-LFA met the WHO TPP acceptable specificity relative to both AN and NP PCR, and the sensitivity target relative to AN PCR. However, the OA-LFA, along with the two commercially available tests, failed to detect the subset of patients with low viral loads, which represented less than one-fifth of patients with positive results in the population sampled. The reader app was able to reproduce visually read results by trained clinical research staff with near-perfect efficiency, and perhaps better in a subset of cases, which could be an important benefit for settings where users have less training.

This study elicits several important insights. One is the importance of the choice of reference standard for evaluating antigen rapid tests for SARS-CoV-2. In this study, NP PCR detected additional cases as compared to AN PCR a majority of those cases had very low viral loads. As described, the extraction methods differed slightly between NP and AN swabs. However, the relative performance is consistent with that reported elsewhere [17, 18]. The authors believe that AN PCR is a sufficient reference assay in the evaluation of rapid tests, where the intended use is to detect the majority of patients with early, acute SARS-CoV-2 infection, allowing for immediate implementation of isolation and other efforts to arrest transmission of the virus. A second item is that the sensitivity will be dependent on the population tested and the distribution of the viral burden. In a different use case, such as screening of asymptomatic patients where viral load may be shifted lower, then we would expect all the rapid tests, including the OA-LFA, to perform worse than reported in this study. Performance at higher viral loads is perhaps most relevant for the use case of detecting infectious individuals who are more likely to transmit the virus [19, 20].

In this study OA-LFA comparison to commercial assays showed similar performance, although pairwise comparison demonstrated the superiority of the BinaxNOW™ format. We speculate this may be related to the innovative form factor of the BinaxNOW™, with minimal sample dilution and high transfer efficiency. Lastly, the reference assay on MSD did not perform any better than rapid antigen tests. We speculate that this may be due to specimen degradation and loss or alteration of target antigen due to freeze-thaw. These hypotheses require further exploration.

A limitation to this study is that participants self-identified with having symptoms within 7 days of self-reported onset. There remains a strong need for serial testing cohort studies among exposed individuals to characterize how rapid test performs over the time patients are infectious. This study also has limited generalizability, because it was conducted in a single population, by trained health care personnel, at a peak epidemic time. A final limitation of this study is the lack of a viral culture reference; this reference would allow us to the opportunity to more directly assess each RDT's ability to detect individuals that pose a significant transmission risk.

The OA-LFA information assay architecture is publicly disseminated as open access. Researchers can replicate it without proprietary research and development on their own. If selected antibodies become unavailable by supply or cost, we have reported alternatives, and perhaps better performers, which can be substituted in the development process [21]. The design can also be submitted to WHO's Covid Technology Access Pool (C-TAP) and shared with manufacturers in Low- and Middle- Income Countries (LMICs) for their consideration. Additional collaboration and technical assistant may be warranted to assist global LFA manufacturing capacity to meet the demand of the pandemic.

## Supporting information

**S1 Fig. (A)** Reference image of cassette used in this study, including the added sticker of ArUco codes. (**B**) Blue circles indicate the keypoints used in the SIFT algorithm for image recognition and transformation. Keypoints were filtered to ensure that pixels in the read window or sample pad were excluded from the matching process.
(TIF)

**S2 Fig. Selected screenshots from the various stages of the user interacting with the reader app. (A)** Attempting to capture an image of the cassette—note the instructions to the user on top to move the phone and then keep it steady, as well as some but not all quality checks being passed. (**B**) Representative view of the cassette passing all quality checks, but just out of flatness range to facilitate screen grab. (**C**) Result window showing the full captured image, the extracted read window, and locations and intensities (peak height of red channel) of control and test lines, if found.
(TIF)

**S3 Fig.** ROC curves for USmellIt relative to NP (**A**) and AN (**B**) PCR. A cut-off value of a USMELLIT score of 0 retains 100% specificity while providing 22% sensitivity for NP PCR while only 92% specificity is seen relative to AN PCR. The combination of an antigen-based test with a score of 0 offers potential improvement over an antigen test alone, as discussed in the main text.
(TIF)

**S1 Table. Results from McNemar's mid-P value test comparing the sensitivity and specificity for all three tests against AN and NP swab PCR.** Additionally, results from the McNemar mid-P test are shown for the sensitivity for cases with viral loads greater than and less than 1000 copies/μL.
(DOCX)

**S2 Table. Sensitivity and specificity of each test relative to AN and NP PCR with or without the inclusion of the zero USMELLIT criterion.**
(DOCX)

**S1 Data.**
(CSV)

**S1 File.**
(DOCX)

## Acknowledgments

The authors would like to acknowledge CureBase and the CovidClinic for their support in setting up and implementing this study. A special thanks to the creators of USMELLIT for their involvement in the study.

## Author Contributions

**Conceptualization:** Christine M. Bachman, Luis F. Alonzo, Spencer Garing, Sam A. Byrnes, Rafael Rivera, Matthew D. Keller, David M. Cate, Kevin P. Nichols, Puneet Dewan.

**Formal analysis:** Benjamin D. Grant, Stephen Burkot.

**Methodology:** Christine M. Bachman, Rafael Rivera.

**Supervision:** Christine M. Bachman.

**Validation:** Luis F. Alonzo, Spencer Garing, Sam A. Byrnes, Alexey Ball, James W. Stafford, Wenbo Wang, Dipayan Banik, Matthew D. Keller.

**Writing – original draft:** Christine M. Bachman, Benjamin D. Grant, Caitlin E. Anderson, Stephen Burkot, Matthew D. Keller, Puneet Dewan.

**Writing – review & editing:** Sam A. Byrnes, James W. Stafford, Wenbo Wang, Dipayan Banik, David M. Cate, Bernhard H. Weigl.

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
