## [Decision Letter · Decision Letter 0]

21 Jul 2021

PONE-D-21-19695

Clinical validation of an open-access SARS-COV-2 antigen detection lateral flow assay, compared to commercially available assays.

PLOS ONE

Dear Dr. Bachman,

Thank you for submitting your manuscript to PLOS ONE. After careful consideration, we feel that it has merit but does not fully meet PLOS ONE’s publication criteria as it currently stands. Therefore, we invite you to submit a revised version of the manuscript that addresses the points raised during the review process.

Please pay special attention to the methods that the reviewers suggested.

We look forward to receiving your revised manuscript.

Kind regards,

Etsuro Ito

Academic Editor

PLOS ONE

3. You indicated that you had ethical approval for your study. In your Methods section, please ensure you have also stated whether the IRB committee approved the format of written consent via a phone app.

5. Please upload a copy of Supporting Information Figure 3 which you refer to in your text on page 13.

Reviewers' comments:

Reviewer's Responses to Questions

**Comments to the Author**

1. Is the manuscript technically sound, and do the data support the conclusions?

Reviewer #1: Yes

Reviewer #2: Yes

Reviewer #3: Yes

2. Has the statistical analysis been performed appropriately and rigorously? 

Reviewer #1: Yes

Reviewer #2: I Don't Know

Reviewer #3: Yes

3. Have the authors made all data underlying the findings in their manuscript fully available?

Reviewer #1: Yes

Reviewer #2: Yes

Reviewer #3: Yes

4. Is the manuscript presented in an intelligible fashion and written in standard English?

Reviewer #1: Yes

Reviewer #2: Yes

Reviewer #3: Yes

5. Review Comments to the Author

Reviewer #1: The study was designed scientifically and the manuscript was prepared following a standard format. The results and conclusions are valuable for the researchers in the field of lateral flow assay and medical technologists as potential end users. I do feel that the data capturing and analyses through ArUco codes are quite new to the researchers in the field and would like to expect the authors to introduce more details separately.

Reviewer #2: This paper reports a prospective diagnostic accuracy study comparing an open access lateral flow antigen detection assay to two different commercially available assays. An evaluation of a mobile phone reader and the addition of a system to evaluate sense of smell. The study is generally well conducted and reported. The sample size is small, particular for precise estimation of specificity, however the results for the open access assay are promising, particularly for manufacturers in low and middle income countries and when the assay is used in combination with the ‘USMELLIT’ device. Much of the supplementary material seems to be missing, e.g. Figure S2 and all of the Supplementary tables, which is unfortunate. I have a few comments that the authors may wish to consider.

Pg 4. Methods, Participants – please provide a definition of ‘symptomatic’, i.e. was one of the listed symptoms sufficient or was more than one required

Pg 4. Methods, Point-of-care test methods – Please state whether the Sofia and BinaxNOW assays are validated for use with AN swabs. Please also describe who collected the swab and carried out the test (self-collected, health care professional etc), whether the NP swab was obtained first or after the 4 AN swabs, and importantly whether the antigen tests were interpreted blinded to the result of the others.

Pg 7. Methods, Reference Method testing – Were those carrying out the PCR testing blinded to the results of the antigen detection tests?

Pg 8, Methods, Mobile reader application – Line 193-196 are difficult to follow for the general reader, especially as the Figure is not accessible. Please consider explaining in less technical language. Also, although the study does not develop the detection algorithm used in the app, modifications to it are made for the study purposes. ‘Pre-trial testing’ is mentioned – this appears to have been used to set the threshold at which the app considers a test line to be present. The supplementary materials should provide further details about this as one would usually expect to see an algorithm (or modification of an existing algorithm) to be trained on one set of images and tested on another independent set. Perhaps the authors could comment on whether this was done and whether one can expect the result observed here to be repeated using different sets of images.

Pg 9, Methods, Analysis – Please use the terms sensitivity and specificity in preference to PPA and NPA as the latter can easily be confused with positive and negative predictive values.

Pg 10 Results, line 241-242 – The threshold chosen for comparing results in higher/lower viral load is quite low – most studies use a threshold of 10^5 or 10^6 copies/ml. The WHO acceptable sensitivity of 80% refers to results in all samples and not restricted to higher viral load

Pg 12 line 262 – It is currently recommended not to rely on the ‘statistical significance’ of comparisons between tests but to point to differences that might be within or beyond that expected by chance. I suspect the study was not powered to detect differences in specificity therefore I would remove the statement about significance (line 264-265)

Pg 13 line 287-288. The empirical data underlying the sensitivity estimates for the antigen tests combined with the USMELLIT should be reported.

Discussion - The authors do not clearly make the case for conducting AN PCR as well as NP PCR. To my knowledge the WHO do not specify a sample type for the reference method against which antigen tests should be compared for establishing performance characteristics however there is no evidence that the cases detected by AN PCR are the more clinically relevant (infectious cases), therefore care should be taken when interpreting these results otherwise it does look a bit like cherry picking the results to make the tests look better.

Reviewer #3: Bacham and colleagues have submitted an article presenting a clinical validation of an open-access lateral flow assay (OA-LFA) design using commercially available materials and reagents, along with RT-qPCR, BinaxNOW® and Sofia® rapid diagnostic tests. They found that the open-access LFA meets the minimum WHO target product profile for a rapid test with a positive predictive agreement with NP sampling of 69% (60% -78%) for OA-LFA as compared to 74% (64% - 82%) for Sofia®, and 82% (73% - 88%) BinaxNOW™ in adults patients with COVID-19 symptoms less than 7 days. Bacham et al include the evaluation of a reader app that could largely mirror visual interpretation of Labs

Test.

There are however few some points to address before being considered for publication in PlosOne

1. The authors have rightly chosen to calculate the positive and negative predictive agreement to assess the performance of the OA-LFA. However, most of the evaluation of diagnostic tests use Sensitivity, specificity, PPV and NPV. We recommend the authors to include those parameters using the RT-qPCR as the reference as done in most rapid diagnostic tests evaluation. This will make easier the comparison with other evaluations. Moreover in line 233-235 the authors refer to the WHO TPP requirement of 80% sensitivity and 97 specificity.

2. In table 1 the authors provide important data to appreciate the performance of each test. It would be better to also provide the 2x2 table for each vs RT-qPCR results. If this is not possible in the main text they could be added in the supplementary

6. PLOS authors have the option to publish the peer review history of their article (what does this mean?). If published, this will include your full peer review and any attached files.

Reviewer #1: **Yes: **Jianfu Jeffrey Wang

Reviewer #2: **Yes: **Jacqueline Dinnes

Reviewer #3: **Yes: **Yap Boum

---

## [Author Response · Author response to Decision Letter 0]

3 Aug 2021

COMPLETE

2. Please review your reference list to ensure that it is complete and correct. If you have cited papers that have been retracted, please include the rationale for doing so in the manuscript text, or remove these references and replace them with relevant current references. Any changes to the reference list should be mentioned in the rebuttal letter that accompanies your revised manuscript. If you need to cite a retracted article, indicate the article’s retracted status in the References list and also include a citation and full reference for the retraction notice. COMPLETE

3. You indicated that you had ethical approval for your study. In your Methods section, please ensure you have also stated whether the IRB committee approved the format of written consent via a phone app. COMPLETE

4. Your ethics statement should only appear in the Methods section of your manuscript. If your ethics statement is written in any section besides the Methods, please move it to the Methods section and delete it from any other section. Please ensure that your ethics statement is included in your manuscript, as the ethics statement entered into the online submission form will not be published alongside your manuscript. COMPLETE 

5. Please upload a copy of Supporting Information Figure 3 which you refer to in your text on page 13. COMPLETE

6. Please include captions for your Supporting Information files at the end of your manuscript, and update any in-text citations to match accordingly. Please see our Supporting Information guidelines for more I think we are good on this – will review at the end. information: http://journals.plos.org/plosone/s/supporting-information.

Reviewers' comments:

Reviewer's Responses to Questions

Comments to the Author

1. Is the manuscript technically sound, and do the data support the conclusions?

Reviewer #1: Yes

Reviewer #2: Yes

Reviewer #3: Yes

2. Has the statistical analysis been performed appropriately and rigorously?

Reviewer #1: Yes

Reviewer #2: I Don't Know

Reviewer #3: Yes

3. Have the authors made all data underlying the findings in their manuscript fully available?

Reviewer #1: Yes

Reviewer #2: Yes

Reviewer #3: Yes

4. Is the manuscript presented in an intelligible fashion and written in standard English?

Reviewer #1: Yes

Reviewer #2: Yes

Reviewer #3: Yes

5. Review Comments to the Author

Reviewer #1: The study was designed scientifically and the manuscript was prepared following a standard format. The results and conclusions are valuable for the researchers in the field of lateral flow assay and medical technologists as potential end users. I do feel that the data capturing and analyses through ArUco codes are quite new to the researchers in the field and would like to expect the authors to introduce more details separately. The details are included in Supplemental Methods and is now more clearly introduced on Page 7.

Reviewer #2: This paper reports a prospective diagnostic accuracy study comparing an open access lateral flow antigen detection assay to two different commercially available assays. An evaluation of a mobile phone reader and the addition of a system to evaluate sense of smell. The study is generally well conducted and reported. The sample size is small, particular for precise estimation of specificity, however the results for the open access assay are promising, particularly for manufacturers in low and middle income countries and when the assay is used in combination with the ‘USMELLIT’ device. Much of the supplementary material seems to be missing, e.g. Figure S2 and all of the Supplementary tables, which is unfortunate. I have a few comments that the authors may wish to consider.

Pg 4. Methods, Participants – please provide a definition of ‘symptomatic’, i.e. was one of the listed symptoms sufficient or was more than one required COMPLETE

Pg 4. Methods, Point-of-care test methods – Please state whether the Sofia and BinaxNOW assays are validated for use with AN swabs. INDICATED on PAGE 5, per following kit instruction. Please also describe who collected the swab and carried out the test (self-collected, health care professional etc), COMPLETE whether the NP swab was obtained first or after the 4 AN swabs, and importantly whether the antigen tests were interpreted blinded to the result of the others. INCLUDED

Pg 7. Methods, Reference Method testing – Were those carrying out the PCR testing blinded to the results of the antigen detection tests? YES, ADDED

Pg 8, Methods, Mobile reader application – Line 193-196 are difficult to follow for the general reader, especially as the Figure is not accessible. Please consider explaining in less technical language. Also, although the study does not develop the detection algorithm used in the app, modifications to it are made for the study purposes. ‘Pre-trial testing’ is mentioned– this appears to have been used to set the threshold at which the app considers a test line to be present. The supplementary materials should provide further details about this as one would usually expect to see an algorithm (or modification of an existing algorithm) to be trained on one set of images and tested on another independent set. Perhaps the authors could comment on whether this was done and whether one can expect the result observed here to be repeated using different sets of images. Further details on the detection of the test line are provided in Supplemental Methods section 1.4, although this portion may have accidentally been excluded by the submission process initially. The portion of the manuscript called out by the reviewer now explicitly states the presence of these additional details and was further modified slightly for clarity. In general, the peak detection algorithm is deterministic so does not need to be “trained” like a machine learning algorithm – rather, it just needs an appropriate peak height cutoff value.

Pg 9, Methods, Analysis – Please use the terms sensitivity and specificity in preference to PPA and NPA as the latter can easily be confused with positive and negative predictive values.

We have switched to sensitivity and specificity as recommended. 

Pg 10 Results, line 241-242 – The threshold chosen for comparing results in higher/lower viral load is quite low – most studies use a threshold of 10^5 or 10^6 copies/ml. The WHO acceptable sensitivity of 80% refers to results in all samples and not restricted to higher viral load 

We have adjusted our cut-off for higher/lower viral load to be 10^6 copies/mL in order to align with the WHO TPP acceptable analytical sensitivity cutoff. Additionally, we agree the acceptable clinical sensitivity refers to all samples. We have edited the manuscript to report on the clinical sensitivity relative to the TPP based on both AN and NP reference PCR swabs, but not based on viral load stratification. 

Pg 12 line 262 – It is currently recommended not to rely on the ‘statistical significance’ of comparisons between tests but to point to differences that might be within or beyond that expected by chance. I suspect the study was not powered to detect differences in specificity therefore I would remove the statement about significance (line 264-265) New language added to this section.

Pg 13 line 287-288. The empirical data underlying the sensitivity estimates for the antigen tests combined with the USMELLIT should be reported. 

The data has been added to supplemental table 2. 

Discussion - The authors do not clearly make the case for conducting AN PCR as well as NP PCR. To my knowledge the WHO do not specify a sample type for the reference method against which antigen tests should be compared for establishing performance characteristics however there is no evidence that the cases detected by AN PCR are the more clinically relevant (infectious cases), therefore care should be taken when interpreting these results otherwise it does look a bit like cherry picking the results to make the tests look better. 

The decision to include analysis both against AN and NP PCR is to reflect the lack of consensus in the appropriate reference standard. As correctly pointed out by the reviewer, the WHO does not specify a reference test. For their FDA EUA submission, Abbott BinaxNOW utilized an AN swab for reference PCR testing. Quidel Sofia used AN swabs for some subjects and NP swabs for others. We reported performance against both reference swabs in this manuscript to highlight the impact of the reference method choice. Ideally, viral culture would be used as a metric to evaluate sensitivity in patients most likely to transmit the disease. 

Reviewer #3: Bacham and colleagues have submitted an article presenting a clinical validation of an open-access lateral flow assay (OA-LFA) design using commercially available materials and reagents, along with RT-qPCR, BinaxNOW® and Sofia® rapid diagnostic tests. They found that the open-access LFA meets the minimum WHO target product profile for a rapid test with a positive predictive agreement with NP sampling of 69% (60% -78%) for OA-LFA as compared to 74% (64% - 82%) for Sofia®, and 82% (73% - 88%) BinaxNOW™ in adults patients with COVID-19 symptoms less than 7 days. Bacham et al include the evaluation of a reader app that could largely mirror visual interpretation of Labs

Test.

There are however few some points to address before being considered for publication in PlosOne

1. The authors have rightly chosen to calculate the positive and negative predictive agreement to assess the performance of the OA-LFA. However, most of the evaluation of diagnostic tests use Sensitivity, specificity, PPV and NPV. We recommend the authors to include those parameters using the RT-qPCR as the reference as done in most rapid diagnostic tests evaluation. This will make easier the comparison with other evaluations. Moreover in line 233-235 the authors refer to the WHO TPP requirement of 80% sensitivity and 97 specificity. 

We have replaced PPA and NPA with sensitivity and specificity, based on the suggestion of reviewer 2 and 3, and to be consistent with the WHO TPP. 

2. In table 1 the authors provide important data to appreciate the performance of each test. It would be better to also provide the 2x2 table for each vs RT-qPCR results. If this is not possible in the main text they could be added in the supplementary Thoughts, Because we have 3 tables on the results, we found this sufficient. We can supply a 2X2 table upon request.

---

## [Editor Report · Decision Letter 1]

5 Aug 2021

Clinical validation of an open-access SARS-COV-2 antigen detection lateral flow assay, compared to commercially available assays.

PONE-D-21-19695R1

Dear Dr. Bachman,

We’re pleased to inform you that your manuscript has been judged scientifically suitable for publication and will be formally accepted for publication once it meets all outstanding technical requirements.

Kind regards,

Etsuro Ito

Academic Editor

PLOS ONE

Additional Editor Comments:

Thank you for your appropriate revision.

---

## [Editor Report · Acceptance letter]

9 Aug 2021

PONE-D-21-19695R1 

Clinical validation of an open-access SARS-COV-2 antigen detection lateral flow assay, compared to commercially available assays. 

Dear Dr. Bachman:

I'm pleased to inform you that your manuscript has been deemed suitable for publication in PLOS ONE. Congratulations! Your manuscript is now with our production department. 

Kind regards, 

on behalf of

Prof. Etsuro Ito 

Academic Editor

PLOS ONE